



# The Geodynamic World Builder: a solution for complex initial conditions in numerical modelling

Menno Fraters[1], Cedric Thieulot[1], Arie van den Berg[1], and Wim Spakman[1,2]

[1] Department of Earth Sciences, Faculty of Geosciences, Utrecht University, Utrecht, the Netherlands
[2]Centre of Earth Evolution and Dynamics (CEED), University of Oslo, Norway

**Correspondence:** Menno Fraters (menno.fraters@outlook.com)

**Abstract.** The Geodynamic World Builder is an open source code library intended to set up initial conditions for computational geodynamic models in both Cartesian and Spherical geometries. The inputs for the JSON-style parameter file are not mathematical, but rather a structured nested list describing tectonic features, e.g. a continental, an oceanic or a subducting plate. Each of these tectonic features can be assigned a specific temperature profile (e.g. plate model) or composition label (e.g. uniform).

For each point in space, the Geodynamic World Builder can return the composition and/or temperature. It is written in C++, but can be used in almost any language through its C and Fortran wrappers. Various examples of 2D and 3D subduction settings are presented. The World builder comes with an extensive online User Manual.

## 1  Introduction

Geodynamic modelling has been used in the past four decades to help us better understand the physical processes of Earth's interior including large-scale mantle convection and plate tectonics, or detailed processes of crustal deformation. Numerical modelling of geodynamic processes involves solving the pertinent partial differential equations (PDEs) of mass, momentum and energy conservation supplemented with rheological laws, material parameters and with an equation of thermodynamic state relating, e.g., density, temperature and pressure (e.g. Gerya, 2010; Schubert et al., 2001). In addition these PDEs must be

constrained by boundary conditions, which can be time-dependent, and by initial conditions which describe the starting model for solving the geodynamic problem at hand. For example, 3D initial models of a geometrically simplified nature are often constructed for modelling of generic subduction evolution using plate boundaries and lithosphere domains that are parallel to the sides of the (rectangular) model domain (e.g. Yamato et al., 2009; Stegman et al., 2010; Brune and Autin, 2013; Schellart and Moresi, 2013; Duretz et al., 2014; Holt et al., 2015; Leng and Gurnis, 2015; Naliboff and Buiter, 2015; Kiraly et al., 2016;

Schellart, 2017). When numerically simulating (regions of) the Earth, geometrically more complex initial models are required, e.g., involving the starting plate-tectonic layout, initial trench geometry and slab shape for use either instantaneous dynamics modelling or as initial model for modelling of subduction evolution (e.g. Alisic et al., 2012; Liu and Stegman, 2011; Jadamec





and Billen, 2010, 2012; Chertova et al., 2014; Billen and Arredondo, 2018; Zhou et al., 2018). Such initial model setups cannot be easily created, adapted, or shared with the community, nor easily transferred to another code. We present in this paper a solution to these problems in the form of an open source code library, the Geodynamic World Builder, which has been designed to be user-friendly, extensible, and portable across different platforms. We present the first stable version of the World Builder

which focuses on creating geometrically complex 3D initial models (geometry, composition, and temperature) consisting of first-order plate tectonic features such as continental and oceanic plates, oceanic ridges and transform faults and 3D lithosphere subduction. These configured initial models are intended to help advance research into simulations instantaneous dynamic modelling and of plate tectonic evolution with a wide range of geometric complexity.

## 2   Geodynamic World Builder Philosophy

### 2.1   User Philosophy

In this section we describe the philosophy of how tectonic features such as plates, ridges, faults and slabs can be parametrized by lines and areas that implicitly define volumes to which temperature and composition can be assigned. A composition is a part of the model that is assigned a particular identifying label and in addition an indicator which is given a value between 0 and 1. This indicator can be used by codes using the GWB output to ascribe physical properties to different model regions.

To minimize user effort, the Geodynamic World Builder (GWB) utilizes a parametrization of 3D structures by 2D coordinate input, by defining their (projected) location on the surface. The GWB can be used to create initial models in Cartesian and spherical geometries.

User input files should be specified in JSON (json.org), which is an internationally standardized language (ISO/IEC 21778). We use a relaxed form of JSON which allows comments, NaN's and tailing commas to improve usability through RapidJSON

(http://rapidjson.org/). The user inputs coordinates and can assign particular properties to features such as 'linear' for a temperature profile, or 'uniform' for the compositional makeup of the plate. Note that only a subset of the options is mentioned in this paper. We refer to the online Geodynamic World Builder Manual (https://geodynamicworldbuilder.github.io) for the complete listing.

The GWB uses a hierarchical overlay of features. This means that features defined first are spatially overlain by features

defined later in places where both overlap. The GWB recognizes two types of features: area features and line features, which will be explained in the following sections. A possible third type of features, point features, will be discussed in section 4.

### 2.1.1   Continental lithosphere plate

A continental plate is an 'area feature' in the GWB and is defined by its surface perimeter and its thickness. The perimeter is specified as a list of points which enclose the continental area. Within the defined volume of the continental plate, the GWB

offers various options for defining temperature values and compositions. For example, a continental plate can be assigned multiple layers of different compositions and a linear geotherm that matches a predefined adiabatic mantle temperature at the





base of the lithosphere. We note that continental lithosphere with a variable thickness is a development for future releases of the GWB, but can be mimicked in the present version by specifying contiguous continental areas with different thickness. Also, continental topography is currently not explicitly implemented, but it can be achieved through a sticky air approach, where air is a composition of varying thickness atop the model (Schmeling et al., 2008; Crameri et al., 2012).

### 2.1.2   Oceanic lithosphere plate

Like the continental plate, the oceanic plate is parametrized as an area feature with a flat surface. We have implemented the 'plate model' (e.g. Fowler, 2005) for assigning an age-dependent temperature to oceanic lithosphere. In section 3.1.2 we will show an example of a ridge-transform system with ridge jumps. The workaround for implementing oceanic bathymetry is the same as for the continental lithosphere plate.

### 2.1.3   The mantle

The upper and lower mantle can also be parametrized as an area feature that starts below the lithosphere or at the surface and is overlain by lithosphere in a later building stage. This allows for defining a upper and lower mantle and to insert specific volumetric structures such as Large Low Shear wave Velocity Provinces (LLSVPs) at the core-mantle boundary. In the present version these mantle features can be assigned a radially uniform, linear or adiabatic temperature profile. Future versions may include laterally varying temperature or compositions, e.g. scaled from seismic tomography models (e.g. Steinberger et al., 2015).

### 2.1.4   A subducting plate

A subducting plate is a 'line feature' in the GWB and is defined by the location of the trench and one or more depth segments each describing a part of the geometry of the subducting slab. They are defined by a length and by thicknesses and dip angles at beginning and end of the slab segment. In sequence, these segments can makeup a smoothly varying slab geometry which can for example flatten in the upper mantle transition zone, or may prescribe a slab entering the lower mantle. Every point in the trench coordinate list defines a vertical section of the subducting plate that may consist of one or several slab segments. Both sections and segments can vary in length, dip angle or thickness. The length of a subducting slab is always computed as the length along the top of the slab so that this can straightforwardly represent the amount of relative plate convergence during a certain period. The dip angle is defined as the angle between the surface and the local plunge of the slab. The dip angle is specified at the start and end point of each depth segment along the vertical section. Dip angles are linearly interpolated along a segment. The overall direction of slab dip can be to either side of the trench and is selected by specifying for each subducting plate an additional point at the surface, the 'dip-point', at the slab dip-side of the trench segment. Slab dip is linearly interpolated between subsequent vertical slab segments. This parametrization allows for constructing smoothly varying 3D slab morphology. Note that it is also possible to give slabs a starting depth to configure detached slabs.



For each point at the surface of the slab the depth and the distance to the trench, as measured along the surface, are available and can be used to assign slab temperatures, e.g., by using the McKenzie (1970) slab temperature model.

### 2.1.5 A fault

To allow for complicated fault shapes (e.g. listric faults), faults are also parametrized as line features. An important difference
between faults and subducting plates is that for subducting plates the trench defines the top of the plate at the plate boundary, while for faults the line feature defines the center of the fault with respect to which a fault thickness can be defined.

### 2.2 Code philosophy

The following design principles define the Geodynamic World Builder:

1. *A single text-based input file centered around plate tectonic terminology*: as explained in Section 2.1. The particular syntax is specified in the online manual and will be illustrated with examples below.

2. *Code-, language-, and platform-independence:* The GWB is designed to be integrated in the different geodynamic codes through a simple interface. The library is written in C++, has official interfaces (wrappers) to C and Fortran and it is possible to call the GWB from the command line. Note that the C wrapper enables calling the GWB from almost any other language like Python and Matlab. The code is continuously tested with every change on the Linux, OSX and Windows operating systems.

3. *Up-to-date user manual and code documentation.* Manual and doxygen http://doxygen.nl/ code documentation provided through https://geodynamicworldbuilder.github.io.

4. *Safe use in parallel codes:* The GWB is split into two phases. The setup phase, encapsulated in the function create_world, is not thread safe but when upon completion the generated "world object" is thread-safe and can be used to query temperature and compositions in parallel.

5. *Readable and extensible code:* Following ASPECT (Kronbichler et al., 2012; Heister et al., 2017) we use a plugin system for different parts of the code. Such plugins enable users to add functionalities such as plate tectonic features or coordinate systems without knowledge of the rest of the code.

6. *Version numbering:* using Semantic Versioning 2.0.0 (https://semver.org). The input file should specify the major version number that must match the version number of the used GWB. Before the release of major version 1, backwards incompatible changes may be made in minor versions, because they will be beta releases. This implies that the input files for major version 0 also must contain the minor version number. All these features help ensuring *reproducibility of results*.



## 3   Using the World Builder

To exemplify input files and to show the capabilities of the Geodynamic World Builder, we show here three 2D examples, and two 3D examples of the GWB visualized through the standalone visualization application. This application creates so-called vtu files which can be visualized by programs like Paraview (paraview.org). Furthermore, we show examples of GWB use with the SEPRAN (van den Berg et al., 2015), ELEFANT (Plunder et al., 2018) and ASPECT codes. The annotated input files to create these models are presented in appendixes A to F and are part of the GWB repository.

### 3.1   Standalone examples

The GWB has an option to create a Paraview file of the GWB input file. This can be useful for model creation or visualization support of presenting geodynamic hypotheses, or for checking the user-designed model prior to using it in a next step, e.g., for creating an initial model for geodynamic modelling.

#### 3.1.1   2D subduction

Here we show two subduction models, one in Cartesian coordinates (Fig. 1) and the same model in spherical (effectively cylindrical) coordinates (Fig. 2), which were created through the input files in appendix A. These input files only differ in the selected coordinate system and whether the supplied coordinates are in meters or in degrees. The model has a 95 km thick oceanic plate of which the top 10 km defines the crust and which turns into a 500 km long subducting slab in the center of the domain. The temperature in the oceanic plate follows the plate model (Fowler, 2005) with a bottom temperature of 1600 K. The slab temperature is computed using the McKenzie model for a particular slab history. The model also contains a 100 km thick continental plate of which the top 30 km is crust. Furthermore, the upper and lower mantle are given different compositions and follow a linear temperature profile in the upper mantle from 1600 K at 95 km depth to 1820 K at 660 km depth, and in the lower mantle from 1820 at 660 km depth to 2000 at 1160 km depth.

This example is created by placing the features in a particular order in the input file. The features overlay, and in this case overwrite, an adiabatic background temperature and all compositions set to zero. This example consists of five features: an oceanic plate, a continental plate, an upper mantle, a lower mantle and a subducting plate. The first four do not overlap in their input definition, so the order of definition in the Geodynamic World Builder input file does not make a difference in the result. The subducting plate overwrites parts of the oceanic plate, continental plate and the upper mantle, which is effectuated by defining the slab after these three features. For each feature temperature and composition models are selected.

#### 3.1.2   3D ocean spreading

We show in figure 3 a 3D rifting model with two rift systems next to each other. The temperature is defined by the plate model. The mantle is given an adiabatic geotherm defined by $\theta_S \exp(\alpha g d/C_p)$, where $\theta_S$ is the potential surface temperature of the mantle, $\alpha$ is the thermal expansion coefficient, $g$ is the gravitational acceleration, $C_p$ is the specific heat and $d$ is the depth. The input file of this example consists of the definition of the mantle domain followed by two oceanic plates, which form the two





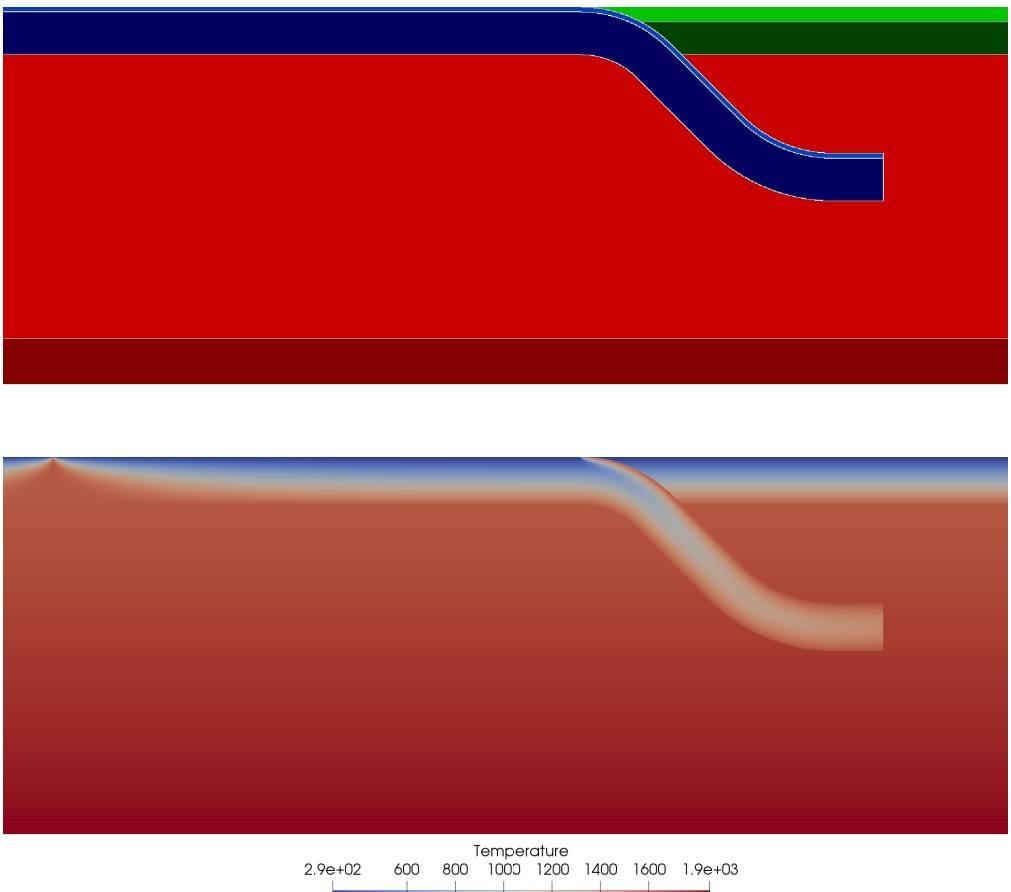

**Figure 1.** The top figure shows the distribution of different compositions through the model domain. The oceanic crust composition is light blue, the oceanic lithosphere is dark blue, the continental crust is light green, the continental lithosphere is dark green, the upper mantle is light red and the lower mantle dark red. The bottom figure shows the temperature (in Kelvin) distribution in the model.

ridge-plate systems. The two oceanic plates are exactly the same, except for the shifted ridge location. The input file for this example can be found in Appendix B.

### 3.1.3 3D subduction

Figure 4 shows a 3D example defining a subduction geometry similar to the one in Plunder et al. (2018). In this example the

5    trench consists of three connected straight lines. To create a smooth transition between these sections, the user can choose to use a monotone spline interpolation between the coordinates given by the user. This example includes a linear temperature upper and lower mantle as described in the 2D subduction example. The 95 km thick oceanic plate and the 120 km thick continental plate features are both defined before the subducting plate feature, of which the trench is defined along the interface between the two. The slab itself is 95 km thick and consists of four segments. One 200 km long segment which goes from a dip angle





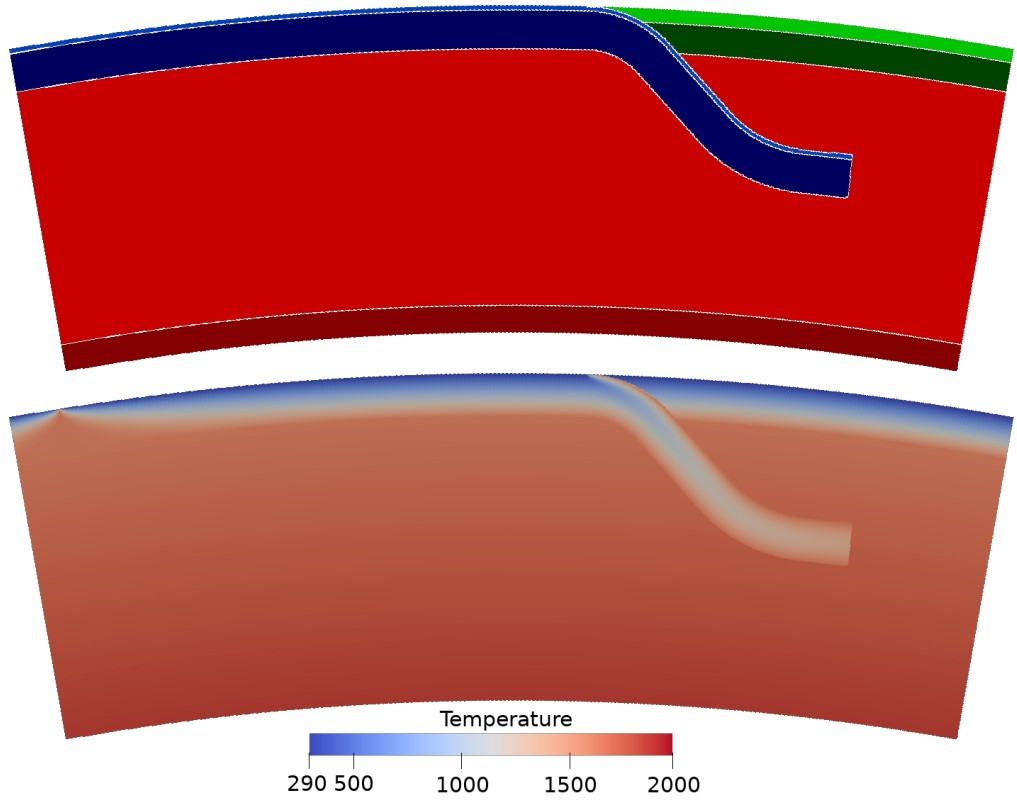

**Figure 2.** The same as setup as in figure 1, but now in spherical geometry. The top figure shows the composition, the bottom figure shows the temperature.

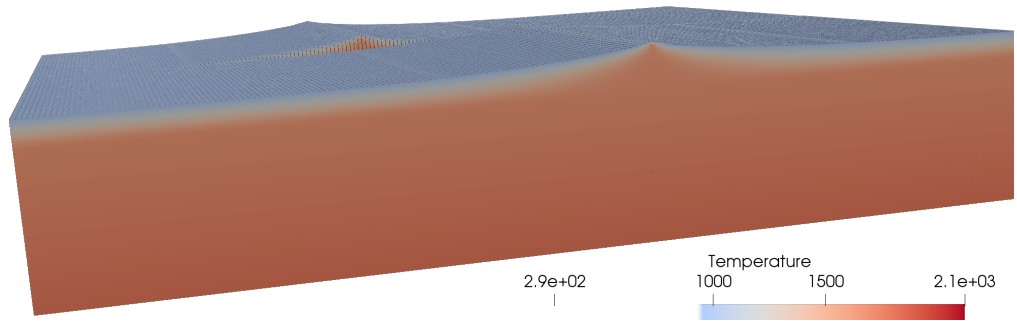

**Figure 3.** The temperature field of the 3D two rift systems example. Material with a temperature below 950 K has been omitted, in order to better show the rifts. Note the second rift system in the background.





**Figure 4.** The temperature field of the 3D subduction example. Note the smooth transition between the upper and lower part of the subduction system in the top figure and the curved geometry of the slab in the lower figure. For visualization purposes we have omitted the top 25 km of the model in the top figure.

of 0° to 45°, and one 400 km long segment which has an angle of 45°, one 200 km long segment which goes from 45° to 0° and one 100 km long segment, with constant dip angle of 0°. The input file for this example can be found in Appendix C.

## 3.2 Using the GWB with SEPRAN

SEPRAN is a general purpose finite element toolkit applied in engineering problems as well as in development of 2D and
5  3D numerical models in geodynamics and planetary science (Chertova et al., 2012, 2014; Čížková et al., 2012; van den Berg et al., 2015, 2019; Zhao et al., 2019). The model contains a lithospheric slab subducting under an overriding plate as shown in





Fig. 5. One sided subduction is obtained in a self consistent way by the presence of a weak crustal layer of uniform viscosity $10^{21}$ Pa·s on top of the subducting lithosphere. The mantle underlying the crust has a temperature and pressure dependent viscosity with an Arrhenius type parametrization representative of diffusion creep in olivine under upper mantle pressure and temperature conditions. Viscosity is modeled as a material property for the crustal layer material and the mantle material. Material transport is implemented using particle tracers that are advected by the convective flow. The medium is described as a mechanical mixture of materials with contrasting properties.

A 2D rectangular domain of 1000 km depth and 2000 km width is used. The initial thermal and composition state is created using the Fortran wrapper of the GWB library. The GWB tool is called in a loop over all nodal points of the FEM mesh to define the initial temperature field for the subsequent convection calculations. In a similar way the material distribution of the initial state is defined by calling the composition function of the GWB library in a program loop over particle tracers. The input file for this example can be found in Appendix D.

### 3.3 Using the GWB with ELEFANT

ELEFANT is a 2D/3D Finite Element code for geodynamic problems (Maffione et al., 2015; Lavecchia et al., 2017; Thieulot, 2017; Plunder et al., 2018) written in Fortran. It principally relies on bi/tri-linear velocity-constant pressure elements and uses the Marker-in-Cell technique to track materials. In order to demonstrate the GWB flexibility of use a 3D double subduction setup was created with the Fortran wrapper of the GWB (see Fig. 6): a composition between 1 and 6 was then easily assigned to all markers (two different oceanic crusts and oceanic lithospheres, one upper mantle and one lower mantle) and a temperature based on the McKenzie model (McKenzie, 1970) was prescribed onto the FE mesh, as shown in Fig. 7.

The domain is a Cartesian box with dimensions $2000 \times 2000 \times 800$ km and the Finite Element mesh counts $120 \times 120 \times 50 = 720,000$ elements. Each element contains 64 randomly distributed markers. Free slip boundary conditions are imposed at the bottom ($z = 0$), top ($z = L_z$) and sides ($y = 0$ and $y = L_y$) of the domain. The other two sides, $x = 0$ and $x = L_x$, are a mix of free slip (for $z < 100$km or $z > 690$km) and open boundary conditions (for $100 < z < 690$ km) (Chertova et al., 2012). The input file for this example can be found in Appendix E.

### 3.4 Using the GWB with ASPECT

ASPECT is an open source community FEM designed for geodynamic problems (Heister et al., 2017; Kronbichler et al., 2012). The model which was run with ASPECT is a 3D Cartesian model of a curved subduction system similar to the plate-tectonic setting of the Lesser Antilles subduction of the eastern Caribbean region. The lithosphere consists of a strong zero velocity Caribbean upper plate, surrounded by an oceanic North American plate to the north and northeast and the oceanic-continental South American plate to the south and southeast. In the model the North American and South American plates move west at a average rate over the past 5 Ma of 1.4 cm/yr relative to the Caribbean plate (Boschman et al., 2014). The Lesser Antilles trench curves around the east and north of the Caribbean plate. To the south, the Caribbean plate is partially decoupled from the South American plate by a 50 km wide weak zone. To the northwest a 250 km wide weak zone, from the western end of the trench to the western edge of the model, partially decouples the North American plate from the Caribbean plate. Below the lithosphere





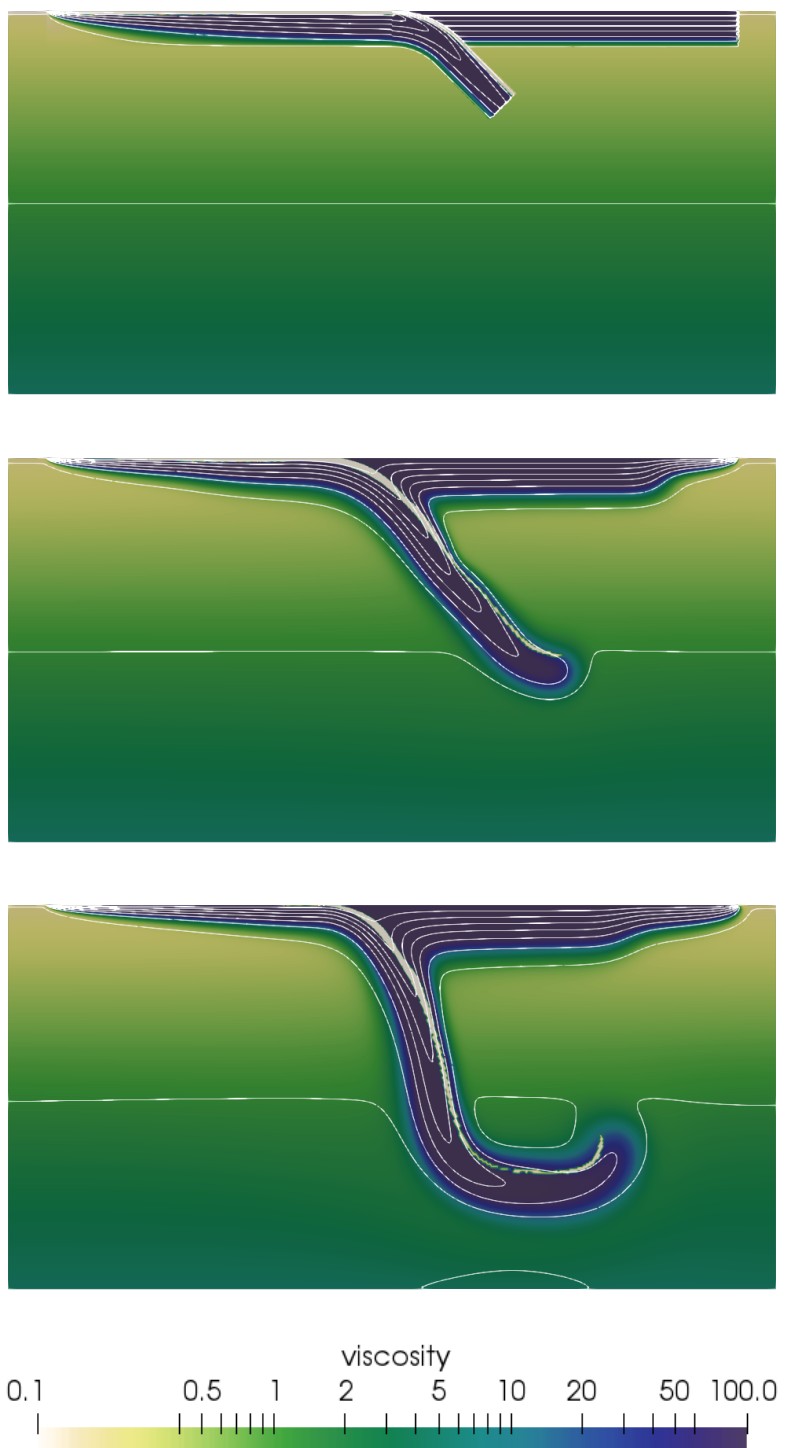

**Figure 5.** Dimensionless viscosity field in log scale superimposed with 10 (dimensionless) temperature (between 0 and 0.82) isocontours.





```
do im=1,nmarker
depth=Lz−zm(im)
do imat=1,nmat
call composition_3d(cworld,xm(im),&
ym(im),zm(im),&
depth,imat−1,flag)
if (flag==imat) then
mat(im)=imat
exit
end if
end do
end do
do ip=1,np
depth=Lz−z(ip)
call temperature_3d(cworld,x(ip),y(ip),&
z(ip),depth,gz,T(ip))
end do
```

**Figure 6.** Example ELEFANT query routine using the GWB supplied Fortran wrappers composition_3d() and temperature_3d(): a) a loop runs over all markers and determines for each the composition at its location; b) a loop runs over all grid points and the GWB returns its temperature as a function of their spatial coordinates.

the sidewalls are open (Chertova et al., 2012, 2014) allowing for horizontal in/out flow of mantle material. From 660 km down a denser and more viscous material has been prescribed to delay sinking of the slab into the lower mantle. The top boundary is a free surface (Rose et al., 2017) and the bottom boundary has a prescribed zero velocity. The result of about 2.5 million years of evolution is shown in figure 8.

5      The details of the setup are presented in Appendix F.

### 3.5   performance

The Finite Element mesh used in the example of section 3.4 is built in several steps by ASPECT: the code starts with a regular grid and allows adaptive mesh refinement to take place one level at the time. Each step of this process calls the GWB library. The first step generates a grid counting 28,000 elements and reports a total setup time for the initial conditions of 3.6 seconds on

10    480 MPI processes. The second step mesh counts 99,000 elements while the setup of the initial conditions took (cumulatively) 10 seconds. The third step sees the number of element jump to about 560,000 elements while its total (cumulative) time to





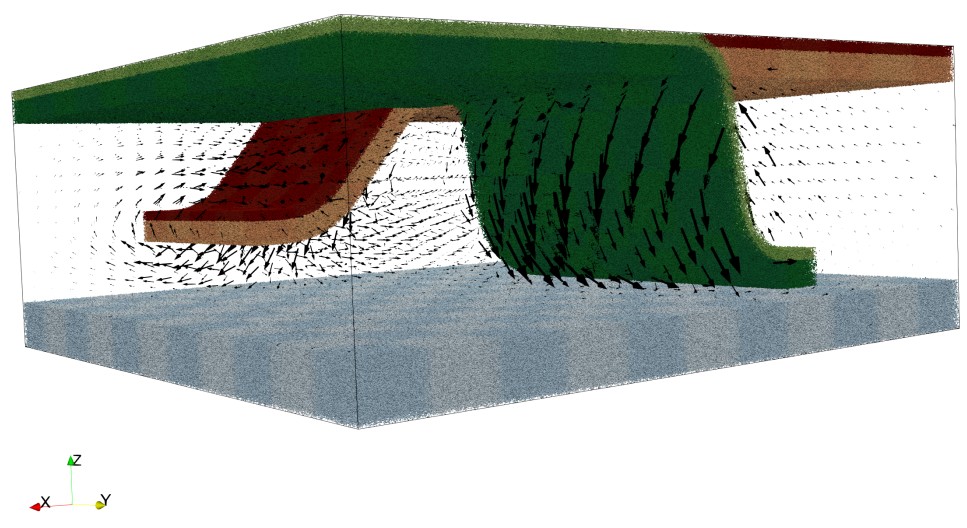

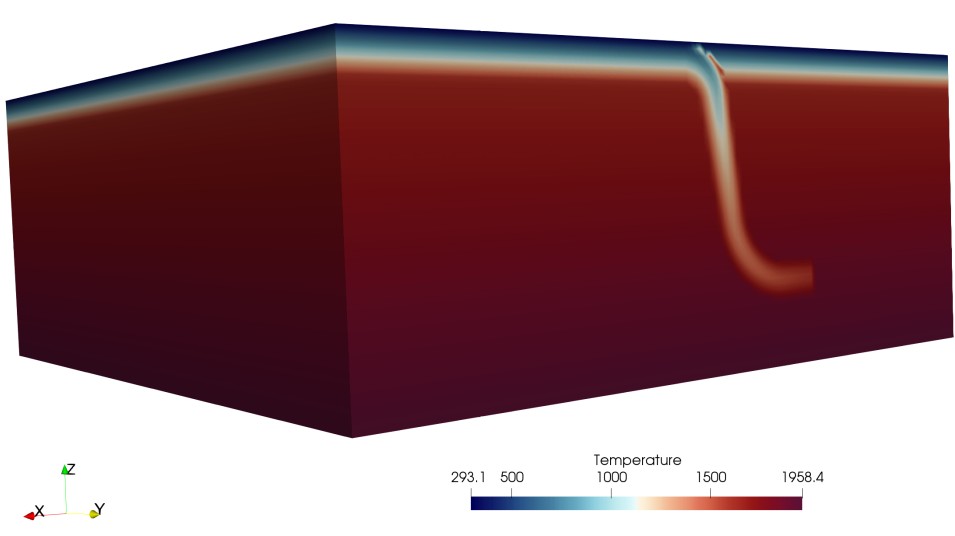

**Figure 7.** Top: Markers for 5 compositions (the mantle markers have been left out for ease of visualization) with the resulting velocity field; Bottom: Temperature field.



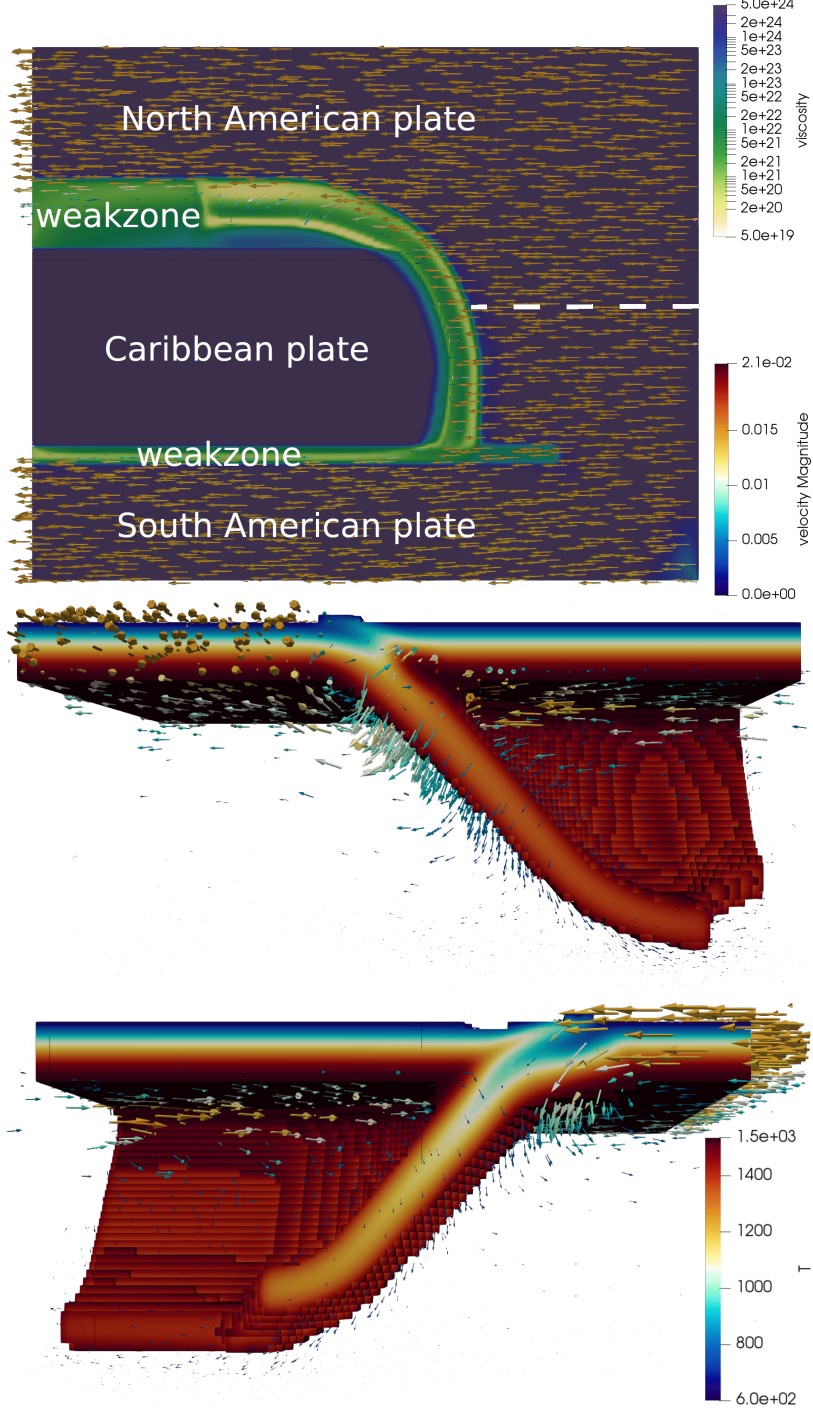

**Figure 8.** The 3D ASPECT Caribbean example after 2.5 million years of evolution. The top image is a top view of the model, where the top 50 km is removed, and where the viscosity field is shown with the velocity field indicated by the arrows. The bottom two figures are cut outs of the temperature field between 600 K and 1535 K, showing in colour the temperature (T) and with arrows the velocity fields, highlighting the velocity field in the slab and lithosphere.



setup the initial conditions remains low at about 36 seconds. This figure represents about 0.7% of the total wall time of the first time step, and a negligible portion of the total wall time of the 20Myr-long simulation.

## 4 Discussion

We presented the Geodynamic World Builder Version 0.2.0 as a tool for constructing 2D and 3D initial models of geodynamic
settings involving crust/lithosphere, plate boundaries, and subduction. The interface of the GWB with a numerical modelling code is based on a query of the modelling code to supply temperature, density, or other information at a particular position. This paper discusses version 0.2.0 of the Geodynamic World Builder, which is considered to be a beta version of the code. Input format and/or functionality may change between minor versions and this will be documented on the website. From version 1.0.0, we will use Semantic Versioning 2.0.0 (https://semver.org/spec/v2.0.0.html), and backwards incompatible changes
will only be made in every major version of the code. Future improvements may for example include extra temperature or composition modules, e.g. derived from tomographic models, new or improved features or even new output interfaces, e.g. velocity boundary conditions or initial topography. As an extension to area and line features, adding point features are another possible improvement to the Geodynamic World Builder. These can represent, for example, a spherical weak seed or a plume. Because of a simple query interface it is in principle possible to use the GWB in connection with existing numerical modelling
codes used by the geodynamic community. The use of the GWB can also just be restricted to creating 2D or 3D geodynamic models/cartoons for, e.g., teaching purposes or for illustrating a complex geodynamic setting.

*Code availability.* The code is freely available at https://geodynamicworldbuilder.github.io under licence LGPLv2.1. All examples presented in this work are available as cookbooks in the code.

*Competing interests.* The authors declare no conflict of interest.

*Acknowledgements.* M.F. acknowledges constructive feedback from the ASPECT community, and especially from T. Heister, W. Bangerth and R. Gassmöller. The authors also acknowledge constructive proofreading by R. Myhill, H. Brett and L. van de Wiel. MF and CT are indebted to the Computational Infrastructure for Geodynamics (CIG) for their recurring participation to the ASPECT hackathons, during which the foundation of this work was laid out. This work is funded by the Netherlands Organization for Scientific Research (NWO), as part of the Caribbean Research program, grant 858.14.070 and partly supported by the Research Council of Norway through its Centres of
Excellence funding scheme, project number 223272. Data visualization is carried out with ParaView software https://paraview.org/.



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





# Appendix A: 2D subduction examples

## A1 Cartesian input file

```
{
"version":"0.2",
"cross section":[[0,0],[100,0]],
"features":
[
6      // defining the oceanic plate
{
"model":"oceanic plate", "name":"oceanic plate",
"coordinates":[[-1e3,-1e3],[1150e3,-1e3],[1150e3,1e3],[-1e3,1e3]],
"temperature models":
[
{"model":"plate model", "max depth":95e3, "bottom temperature":1600,
"spreading velocity":0.005,
"ridge coordinates":[[100e3,-1e3],[100e3,1e3]]}
],
"composition models":
[
{"model":"uniform", "compositions":[0], "max depth":10e3},
{"model":"uniform", "compositions":[1], "min depth":10e3,
"max depth":95e3}
]
},
23     // defining a continental plate
{
"model":"continental plate", "name":"continental plate",
"coordinates":[[1150e3,-1e3],[2001e3,-1e3],[2001e3,1e3],[1150e3,1e3]],
"temperature models":
[
{"model":"linear", "max depth":95e3, "bottom temperature":1600}
],
"composition models":
[
{"model":"uniform", "compositions":[2], "max depth":30e3},
{"model":"uniform", "compositions":[3], "min depth":30e3,
"max depth":65e3}
]
},
38     // defining the upper mantle
{
"model":"mantle layer", "name":"upper mantle",
"min depth":95e3, "max depth":660e3,
"coordinates":[[-1e3,-1e3],[2001e3,-1e3],[2001e3,1e3],[-1e3,1e3]],
"temperature models":
[
{"model":"linear", "min depth":95e3, "max depth":660e3,
"top temperature":1600, "bottom temperature":1820}
],
"composition models":[{"model":"uniform", "compositions":[4]}]
},
50     // defining the lower mantle
{
"model":"mantle layer", "name":"lower mantle",
"min depth":660e3, "max depth":1160e3,
"coordinates":[[-1e3,-1e3],[2001e3,-1e3],[2001e3,1e3],[-1e3,1e3]],
"temperature models":
[
{"model":"linear", "min depth":660e3, "max depth":1160e3,
"top temperature":1820, "bottom temperature":2000}
],
"composition models":[{"model":"uniform", "compositions":[5]}]
},
62     // defining the subducting plate dipping towards the continental plate
{
```

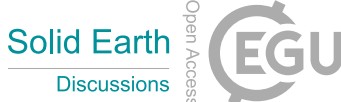

```
"model":"subducting plate", "name":"Subducting plate",
"coordinates":[[1150e3,-1e3],[1150e3,1e3]], "dip point":[2000e3,0],
"segments":[{"length":200e3, "thickness":[95e3], "angle":[0,45]},
{"length":200e3, "thickness":[95e3], "angle":[45]},
{"length":200e3, "thickness":[95e3], "angle":[45,0]},
{"length":100e3, "thickness":[95e3], "angle":[0]}],
"temperature models":
[
{"model":"plate model", "density":3300, "plate velocity":0.01 }
],
"composition models":
[
{"model":"uniform", "compositions":[0], "max distance slab top":10e3},
{"model":"uniform", "compositions":[1], "min distance slab top":10e3,
"max distance slab top":95e3 }
]
}
]
}
```

**Listing 1.** 2D Cartesian subduction example. The lines of green text (preceded by the double forward slashes) are comments and have no effect on the result.

## A2   Spherical input file

```
{
"version":"0.2",
"cross section":[[0,0],[10,0]],
"features":
[
6      // defining the oceanic plate
{
"model":"oceanic plate", "name":"oceanic plate",
"coordinates":[[-1,-1],[11.5,-1],[11.5,1],[-1,1]],
"temperature models":
[
{"model":"plate model", "max depth":95e3, "bottom temperature":1600,
"spreading velocity":0.005,
"ridge coordinates":[[1,-1],[1,1]]}
],
"composition models":
[
{"model":"uniform", "compositions":[0], "max depth":10e3},
{"model":"uniform", "compositions":[1], "min depth":10e3,
"max depth":95e3}
]
},
23     // defining a continental plate
{
"model":"continental plate", "name":"continental plate",
"coordinates":[[11.5,-1],[21,-1],[21,1],[11.5,1]],
"temperature models":
[
{"model":"linear", "max depth":95e3, "bottom temperature":1600}
],
"composition models":
[
{"model":"uniform", "compositions":[2], "max depth":30e3},
{"model":"uniform", "compositions":[3], "min depth":30e3,
"max depth":65e3}
]
},
38     // defining the upper mantle
{
"model":"mantle layer", "name":"upper mantle",
"min depth":95e3, "max depth":660e3,
"coordinates":[[-1,-1],[21,-1],[21,1],[-1,1]],
```



```
"temperature models":
[
{"model":"linear", "min depth":95e3, "max depth":660e3,
"top temperature":1600, "bottom temperature":1820}
],
"composition models":[{"model":"uniform", "compositions":[4]}]
},
50      // defining the lower mantle
{
"model":"mantle layer", "name":"lower mantle",
"min depth":660e3, "max depth":1160e3,
"coordinates":[[-1,-1],[21,-1],[21,1],[-1,1]],
"temperature models":
[
{"model":"linear", "min depth":660e3, "max depth":1160e3,
"top temperature":1820, "bottom temperature":2000}
],
"composition models":[{"model":"uniform", "compositions":[5]}]
},
62      // defining the subducting plate dipping towards the continental plate
{
"model":"subducting plate", "name":"Subducting plate",
"coordinates":[[[11.5,-1],[11.5,1]], "dip point":[20,0],
"segments":[{"length":200e3, "thickness":[95e3], "angle":[0,45]},
{"length":200e3, "thickness":[95e3], "angle":[45]},
{"length":200e3, "thickness":[95e3], "angle":[45,0]},
{"length":100e3, "thickness":[95e3], "angle":[0]}],
"temperature models":
{"model":"plate model", "density":3300, "plate velocity":0.01 }
],
"composition models":
[
{"model":"uniform", "compositions":[0], "max distance slab top":10e3},
{"model":"uniform", "compositions":[1], "min distance slab top":10e3,
"max distance slab top":95e3 }
]
}
]
}
```

**Listing 2.** 2D Spherical subduction example. The lines of green text (preceded by the double forward slashes) are comments and have no effect on the result.

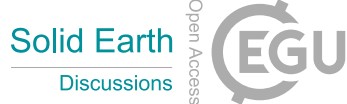



## Appendix B: 3D ocean spreading example input file

```
{
"version":"0.2",
"features":
[
5       // defining one of the oceanic plates with a ridge
{
"model":"oceanic plate", "name":"oceanic plate A",
"coordinates":[[-1e3,-1e3],[2001e3,-1e3],[2001e3,1000e3],[-1e3,1000e3]],
"temperature models":
[
{
"model":"plate model", "max depth":95e3, "spreading velocity":0.005,
"ridge coordinates":[[1200e3,-1e3],[1200e3,1000e3]]
}
],
"composition models":
[
{"model":"uniform", "compositions":[0], "max depth":10e3},
{"model":"uniform", "compositions":[1], "min depth":10e3,
20           "max depth":95e3}]
21      },
22      // defining the other oceanic plate with a ridge
23      {
25        "model":"oceanic plate", "name":"oceanic plate B",
"coordinates":[[-1e3,1000e3],[2001e3,1000e3],[2001e3,2001e3],[-1e3,2001e3]],
"temperature models":
[
{
"model":"plate model", "max depth":95e3, "spreading velocity":0.005,
"ridge coordinates":[[800e3,1000e3],[800e3,2000e3]]
}
],
"composition models":
[
{"model":"uniform", "compositions":[0], "max depth":10e3},
{"model":"uniform", "compositions":[1], "min depth":10e3,
"max depth":95e3}]]
]
}
```

**Listing 3.** 3d ocean spreading example input file. The lines of green text (preceded by the double forward slashes) are comments and have no effect on the result.

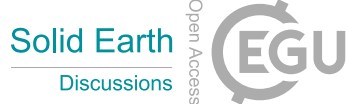

## Appendix C: 3D subduction example input file

```
{
"version":"0.2",
"coordinate system":{"model":"spherical", "depth method":"begin segment"},
"cross section":[[0,0],[10,0]],
"maximum distance between coordinates":0.01,
"interpolation":"monotone spline",
"features":
[
9       // defining the upper mantle
{
"model":"mantle layer", "name":"upper mantle",
"min depth":95e3, "max depth":660e3,
"coordinates":[[-1,-1],[41,-1],[41,-1],[-1,-1]],
"temperature models":
[
{
"model":"linear", "min depth":95e3, "max depth":660e3,
"top temperature":1600, "bottom temperature":1820
}
],
"composition models":[{"model":"uniform", "compositions":[4]}]
},
23      // defining the lower mantle layer
{
"model":"mantle layer", "name":"lower mantle",
"min depth":660e3, "max depth":1160e3,
"coordinates":[[-1,-1],[41,-1],[41,-1],[-1,-1]],
"temperature models":
[
{
"model":"linear", "min depth":660e3, "max depth":1160e3,
"top temperature":1820, "bottom temperature":2000
}
],
"composition models":[{"model":"uniform", "compositions":[5]}]
},
37      // defining the oceanic plate
{
"model":"oceanic plate", "name":"oceanic plate",
"coordinates":[[-1,-1],[-1,41],[15,41],[15,20],[5,10],[5,-1]],
"temperature models":
[{"model":"linear", "max depth":95e3, "bottom temperature":1600}],
"composition models":
[
{"model":"uniform", "compositions":[0], "max depth":10e3},
{"model":"uniform", "compositions":[1], "min depth":10e3,
"max depth":95e3}
]
},
50      // defining the continental plate
{
"model":"continental plate", "name":"continental plate",
"coordinates":[[41,41],[15,41],[15,20],[5,10],[5,-1],[41,-1]],
"temperature models":[{"model":"linear", "max depth":120e3,
"bottom temperature":1600}],
"composition models":
[
{"model":"uniform","compositions":[2], "max depth":30e3},
{"model":"uniform","compositions":[3], "min depth":30e3,
"max depth":120e3}
]
},
63      // defining the subducting plate
{
"model":"subducting plate", "name":"Subducting plate",
"coordinates":[[15,41],[15,25],[5,5],[5,-1]], "dip point":[20,0],
"segments":[{"length":200e3, "thickness":[95e3], "angle":[0,45]},
```



```
{"length":400e3, "thickness":[95e3], "angle":[45]},
{"length":200e3, "thickness":[95e3], "angle":[45,0]},
{"length":100e3, "thickness":[95e3], "angle":[0]}],
"temperature models":
5 72       [{"model":"plate model", "density":3300, "plate velocity":0.05 }],
"composition models":
[
{"model":"uniform", "compositions":[0], "max distance slab top":10e3},
{"model":"uniform", "compositions":[1], "min distance slab top":10e3}
10 77        ]
}
]
}
```

**Listing 4.** 3d subduction spreading example input file. The lines of green text (preceded by the double forward slashes) are comments and have no effect on the result.

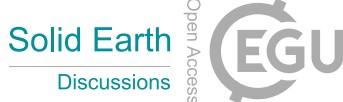

## Appendix D: SEPRAN 2D subduction

```
{
"version":"0.2",
"cross section":[[0,0],[100,0]],
"features":
[
6      // defining an oceanic plate on the left side of the model
{
"model":"oceanic plate", "name":"oceanic plate", "max depth":95e3,
"coordinates":[[-1e3,-1e3],[1000e3,-1e3],[1000e3,1e3],[-1e3,1e3]],
"temperature models":
[
{
"model":"plate model", "max depth":95e3, "bottom temperature":1600,
"spreading velocity":0.01,
"ridge coordinates":[[100e3,-1e3],[0e3,1e3]]
}
],
"composition models":
[
{"model":"uniform", "compositions":[0], "max depth":10e3}
]
},
23     // defining a weakzone oceanic plate at the first 100 km
{
"model":"oceanic plate", "name":"weak zone left", "max depth":95e3,
"coordinates":[[-1e3,-1e3],[100e3,-1e3],[100e3,1e3],[-1e3,1e3]],
"temperature models":
[
{
"model":"linear", "max depth":95e3, "bottom temperature":1600,
"top temperature":1573
}
]
},
35     // defining a continental plate at the right side of the model
{
"model":"continental plate", "name":"continental plate", "max depth":95e3,
"coordinates":[[1000e3,-1e3],[2001e3,-1e3],[2001e3,1e3],[1000e3,1e3]],
"temperature models":
[
{"model":"linear", "max depth":95e3, "bottom temperature":1600}
]
},
44     // defining an oceanic plate as weakzone at the rightmost side of the model
{
"model":"oceanic plate", "name":"weak zone right", "max depth":95e3,
"coordinates":[[1900e3,-1e3],[2000e3,-1e3],[2000e3,1e3],[1900e3,1e3]],
"temperature models":
[
{
"model":"linear", "max depth":95e3, "bottom temperature":1600,
"top temperature":1573
}
]
},
56     // defining the upper mantle
{
"model":"mantle layer", "name":"upper mantle",
"min depth":95e3, "max depth":660e3,
"coordinates":[[-1e3,-1e3],[2001e3,-1e3],[2001e3,1e3],[-1e3,1e3]],
"temperature models":
[
{"model":"linear", "max depth":660e3,
"top temperature":1600, "bottom temperature":1820}
]
},
67     // defining the lower mantle
```

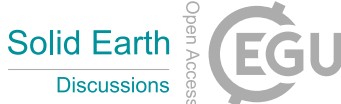


```
{
"model":"mantle layer", "name":"lower mantle",
"min depth":660e3, "max depth":1160e3,
"coordinates":[[-1e3,-1e3],[2001e3,-1e3],[2001e3,1e3],[-1e3,1e3]],
"temperature models":
[
{"model":"linear", "max depth":1160e3,
"top temperature":1820, "bottom temperature":2000}
]
},
78      // defining the subducting plate
{
"model":"subducting plate", "name":"Subducting plate",
"coordinates":[[1000e3,-1e3],[1000e3,1e3]], "dip point":[2000e3,0],
"segments":
[
{"length":200e3, "thickness":[95e3], "angle":[0,45]},
{"length":200e3, "thickness":[95e3], "angle":[45]}
],
"temperature models":
[
{"model":"plate model", "density":3300, "plate velocity":0.01 }
],
"composition models":
[
{"model":"uniform", "compositions":[0], "max distance slab top":10e3}
]
},
96      // defining a continental plate on top of the slab to force 293.15 K at
97      // the surface near the slab
{
"model":"continental plate", "name":"top on slab", "max depth":1,
"coordinates":[[900e3,-1e3],[1100e3,-1e3],[1100e3,1e3],[900e3,1e3]],
"temperature models":[{"model":"uniform", "temperature":293.15}]
}
]
}
```

**Listing 5.** 2d SEPRAN subduction example input file. The lines of green text (preceded by the double forward slashes) are comments and have no effect on the result.





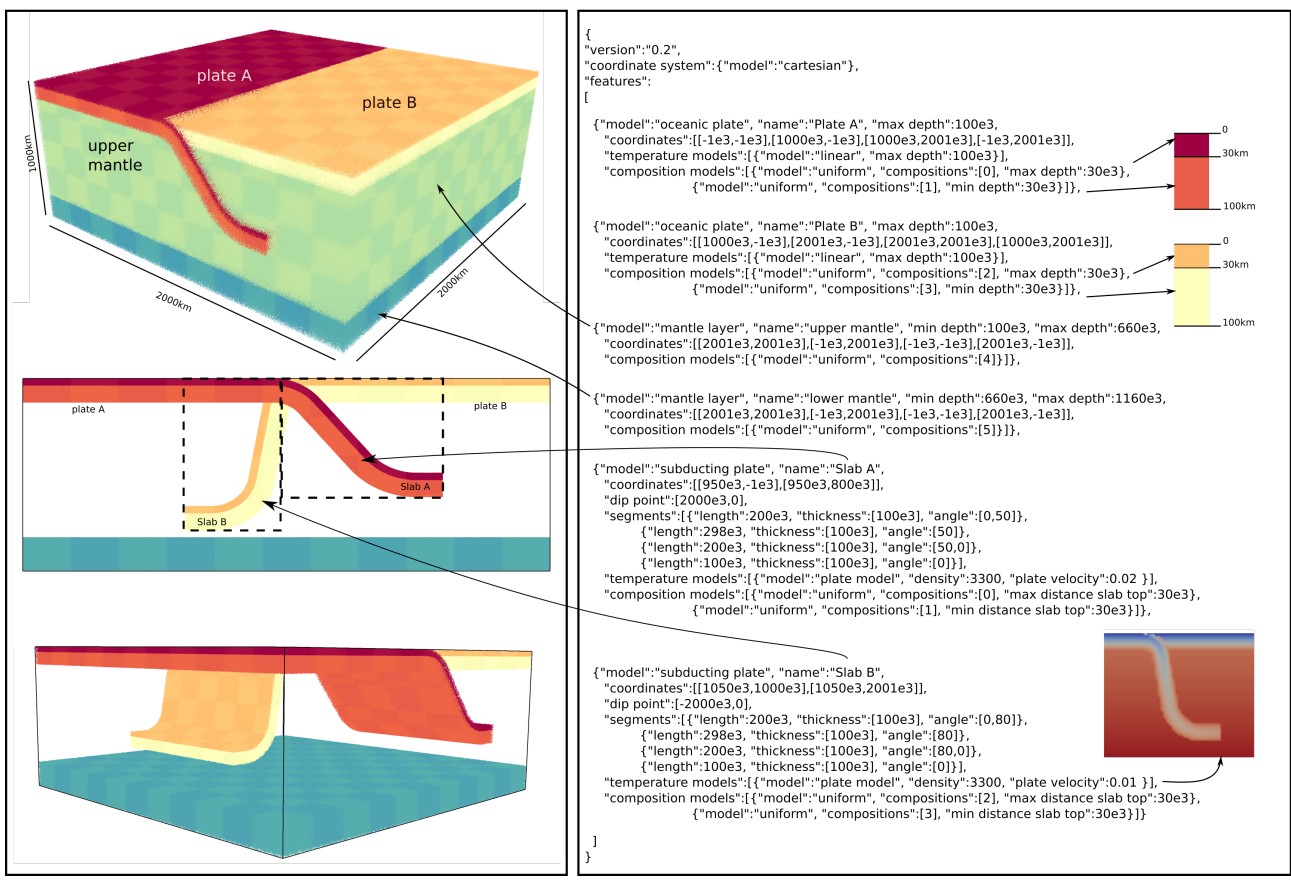

**Figure E1.** Connection between the GWB input file (right panel) and the resulting marker fields (left panel). Small upper inserts in the right panel show each plate layering while the bottom insert shows the temperature field zoomed in on slab B.

## Appendix E: ELEFANT 3D Double subduction setup

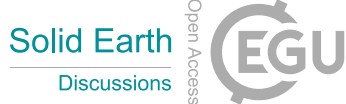

## Appendix F: ASPECT 3d curved subduction

```
{
"version":"0.2",
"potential mantle temperature":1500,
"thermal expansion coefficient":2.0e-5,
"maximum distance between coordinates":100000,
"interpolation":"monotone spline",
"surface temperature":293.15,
"force surface temperature":true,
"coordinate system":{"model":"cartesian"},
"features":
[
12      // defining an oceanic plate for the North and South American plate
{"model":"oceanic plate", "name":"NS American plate",
"coordinates":[[1700e3,0],[1700e3,300e3],[1606e3,650e3],
[1350e3,906e3],[1000e3,1000e3],[-1e3,1000e3],
[-1e3,1501e3],[2501e3,1501e3],[2501e3,-501e3],
[-1e3,-501e3],[-1e3,-50e3],[2000e3,-50e3],
[2000e3,0e3]],
"temperature models":[{"model":"linear", "max depth":100e3}],
"composition models":[{"model":"uniform", "compositions":[0],
"max depth":30e3}]},
23      // Defining an oceanic plate for the Caribbean plate
{"model":"oceanic plate", "name":"Caribbean plate",
"coordinates":[[1700e3,300e3],[1689e3,422e3],[1658e3,539e3],
[1606e3,650e3],[1536e3,749e3],[1450e3,836e3],
[1350e3,906e3],[1239e3,958e3],[1122e3,989e3],
[1000e3,1000e3],[650e3,1000e3],[-1e3,1000e3],
[-1e3,0e3],[1700e3,0e3]],
"temperature models":[{"model":"linear", "max depth":100e3}],
"composition models":[{"model":"uniform", "compositions":[1],
"max depth":30e3}]},
34      // Defining a continental plate for the weak zone
{"model":"continental plate", "name":"Carribean weak zone",
"coordinates":[[-1e3,1000e3],[-1e3,750e3],[1536e3,749e3],
[1450e3,836e3],[1350e3,906e3],[1239e3,958e3],
[1122e3,989e3],[1000e3,1000e3],[650e3,1000e3]],
"temperature models":[{"model":"linear", "max depth":100e3}],
"composition models":[{"model":"uniform", "compositions":[2],
"max depth":30e3},
{"model":"uniform", "compositions":[3],
"min depth":30e3}]},
45      // Defining a mantle layer for the lower mantle
{"model":"mantle layer", "name":"660", "min depth":660e3,
"coordinates":[[-1e3,-500e3],[-501e3,2500e3],[2501e3,2500e3],
[2501e3,-501e3]],
"composition models":[{"model":"uniform", "compositions":[4]}]},
51      // Defining a subducting plate for the Lesser Antilles slab
{"model":"subducting plate", "name":"Lesser Antilles slab",
"coordinates":[[1700e3,0],[1700e3,300e3],[1606e3,650e3],
[1350e3,906e3],[1000e3,1000e3],[650e3,1000e3]],
"dip point":[-1,-1],
"min depth":0, "max depth":660e3,
"segments":
[
{"length":300e3, "thickness":[100e3], "angle":[0,50]},
{"length":371e3, "thickness":[100e3], "angle":[50]},
{"length":275e3, "thickness":[100e3], "angle":[50,0]},
{"length":0e3, "thickness":[100e3], "angle":[0]}
],
"sections":
[
{"coorindate":"0",
"segments":
```





```
[
{"length":300e3, "thickness":[100e3], "angle":[0,25]},
{"length":371e3, "thickness":[100e3], "angle":[50]},
{"length":300e3, "thickness":[100e3], "angle":[50,0]},
{"length":50, "thickness":[100e3], "angle":[0]}
]
},
{"coorindate":"5",
"segments":
[
{"length":300e3, "thickness":[100e3], "angle":[0,25]},
{"length":371e3, "thickness":[100e3], "angle":[50]},
{"length":50e3, "thickness":[100e3], "angle":[50,0]},
{"length":0, "thickness":[100e3], "angle":[0]}
]
}
],
"temperature models":
[
{"model":"plate model", "density":3300, "plate velocity":0.0144,
"thermal conductivity":2.5, "thermal expansion coefficient":2e-5 }
],
"composition models":
[
{"model":"uniform","compositions":[0], "min distance slab top":30e3}
]
},
95    // Defining a continental plate for the weakzone between the Caribbean and
96    // South America
{"model":"continental plate","name":"South Weakzone",
"coordinates":[[-1e3,0e3],[-1e3,-50e3],[2000e3,-50e3],[2000e3,0e3]],
"temperature models":[{"model":"linear", "max depth":100e3}],
"composition models":
[
{"model":"uniform","compositions":[2], "max depth":30e3},
{"model":"uniform", "compositions":[3], "min depth":30e3}]}
]
}
```

**Listing 6.** Input for the ASPECT example. The lines of green text (preceded by the double forward slashes) are comments and have no effect on the result.