# Peer review of "The Geodynamic World Builder: a solution for complex initial conditions in numerical modelling"

_Solid Earth, 2019_

## Referee Comment (RC1) · Fabio A. Capitanio (Referee) · 19 Mar 2019

This paper introduces an interesting tool to support the modelling of complex geometries, temperature distributions and, more in general, initial conditions to numerical models. The paper explains the approach to the problem and the algorithm choice and the coding strategy, then proceed to illustrates some workflows to embed the tool in existing geodynamics computational frames. I find this tool very helpful and timely and I am confident our community could benefit from using it. I have recommendations the Authors could consider, listed below. The paper reads well and I will not comment on the form, yet I would suggest some more explanation in the content: while i understand that the ropes are likely explained in the documentation, a minimum amount of information should be provided here to warrant a separated scientific paper, unless this is

intended as a technical report.

More information is needed on the thermal structure of the slabs. I understand this is derived from McKenzie, 1970, yet I'd recommend the equation solved is presented - briefly - in the paper. This would help the reader understand some critical aspects, such as the velocity assumed to advect the temperature field, if any, whether this solution allows for diffusion or not. This is not clear from figure 1, which seems a bit odd. In principle, this is not a problem, since energy equations routinely solve for both advection and diffusion, in the simplest formulation, yet, strong temperature (and, hence, viscosity) gradients affect the performance of some solvers (mg, for instance) as well as some mesh topologies (e.g., mesh refinements). Some information might help the seamless integration of this tool in the numerical codes. Additionally, this comment applies to the offset ridge in figure 3, where strong horizontal temperature gradient result along the offset zones. More in general, the way lateral variations are handled remains a bit unclear. In principle, one could use blocks where the properties are piecewise constant, yet, this is hardly a natural case and might prompt unwanted inaccuracy in the numerical solution. Perhaps a simple "smoothing" parameter could be considered, as opposed to a proper solution of the (temperature/concentration/material) diffusion equation, which can be done numerically by the preferred code.

I find the treatment of the tectonic provinces appealing, although I'd recommend some more focus on the subduction zone. From my experience, in both the mechanical and thermo-mechanical approaches, a control on the interface is critical. While non-linearities in the upper plate do reduce the coupling along this interface, this might not happen at the inception of subduction, where the available pull forces are low, whence the use of "weak zones" to ease subduction. Indeed, this might not be the case of the model in fig.5, yet in three-dimension, where computational cost is at a premium, allowing alternative strategies is important. Likely a back arc zone can be imposed with a thinned area with a piecewise constant thickness, although a more flexible strategy is in order, here.
Last, but not least. I find that this tool's great potential likely resides in the embedding of realistic datasets. Perhaps the Authors have some example or some idea on workflows using datasets, such as slab 2.0, populating the thermal field in a slab defined by the Benioff zone, or perhaps embedding oceanic lithosphere age dataset into the ridge model, for instance.

---

## Referee Comment (RC2) · Anonymous Referee #2 · 24 Mar 2019

This paper presents a tool, the Geodynamic World Builder, allowing the simulation of more realistic tectonic features, e.g., a continental, an oceanic or a subducting plate. The paper explains the philosophy of the tool and the definition of different tectonic settings. The second part focusses on 2D/3D cartesian and spherical examples of mid-ocean ridges and subduction zones models. I appreciate the effort made to simulate different tectonic context. I have no doubt this tool will be very beneficial and readily usable by the community. The GitHub documentation is also abundant, including a documentation section and a manual.

General comments:

- Basics information, which could benefit potential users, are missing. The models' parameters are not specified, such as the name of the codes used to generate the

figures. - The methodology to define complex geometries/polygons could be more explained. For example, figure E1 is useful and could be included in the main text. - The authors didn't discuss the impact of more realistic geometries on the computational time of the models. Indeed, rapid temperature variation between different materials can induce longer calculation times.

Specific comments:

P1, l15: "...constrained by boundary conditions, which can be time-dependent, and by initial conditions..." you should briefly explain these conditions here and consider adding the parameters of all the models you present in the paper.

P2, l12: "that implicitly define volumes to which temperature and composition can be assigned." Rough variations of temperature, so viscosity, are hard to solve: Could you add a function to avoid this issue?

P3, l3-4: "but it can be achieved through a sticky air approach, where air is a composition...". Yes for small models, but such an approach is difficult to implement in 3D spherical models because it drastically increases the calculation time.

P3, l12-14: "This allows for defining a upper and lower mantle and to insert specific volumetric structures such as Large Low Shear wave Velocity Provinces (LLSVPs) at the core-mantle boundary. In the present version these mantle features can be assigned a radially uniform, linear or adiabatic temperature profile." Could you give an example, it is not clear how you can generate such structures?

P3, l26-30: "Dip angles are linearly interpolated along a segment. The overall direction of slab dip can be to either side of the trench and is selected.... varying 3D slab morphology." A figure, like figure E1, could help the reader to understand the method.

P5, paragraph 3: I encourage the authors to focus on open source software such as CitcomS, CitcomCU, Underworld,....

P5, l17: "The slab temperature is computed using the McKenzie model for a particular

slab history." I understand this paper is not on geodynamic interpretations, but it could help the reader to add the model parameters.

P9, l1-2/Fig. 5: "One sided subduction is obtained in a self-consistent way by the presence of a weak crustal layer of uniform viscosity 1021 PaÅů́s on top of the subducting lithosphere." Is it a self-consistent slab?

P8-9, Paragraph 3.2 to 3.4: the same comment than before: In order to foster the development of open source tools, it could be relevant to add open source software such as CitcomS, CitcomCU, Underworld,..

---

## Author Comment (AC1) · 10 Jul 2019

We would like to express our gratitude to the reviewer for the very careful reading of our manuscript. Peer review makes scientific articles better, and this applies to this one as well – thank you!

The revised paper has been added as a supplement.

*[1] This paper introduces an interesting tool to support the modelling of complex geometries, temperature distributions and, more in general, initial conditions to numerical models. The paper explains the approach to the problem and the algorithm choice and the coding strategy, then proceed to illustrates some workflows to embed the tool in*

[Figure]

*existing geodynamics computational frames. I find this tool very helpful and timely and I am confident our community could benefit from using it. I have recommendations the Authors could consider, listed below. The paper reads well and I will not comment on the form, yet I would suggest some more explanation in the content: while I understand that the ropes are likely explained in the documentation, a minimum amount of information should be provided here to warrant a separated scientific paper, unless this is intended as a technical report.*

Thank you for your kind words. We will be happy to expand the explanations as suggested on the methods used in the paper or in appendices. In general when writing the paper we have chosen to keep the focus of the paper on explaining the concepts instead of implementation details, since the exact details of implementation may change over time, while the proposed concepts should stay the same.

*[2] More information is needed on the thermal structure of the slabs. I understand this is derived from McKenzie, 1970, yet I'd recommend the equation solved is presented - briefly - in the paper. This would help the reader understand some critical aspects, such as the velocity assumed to advect the temperature field, if any, whether this solution allows for diffusion or not. This is not clear from figure 1, which seems a bit odd. In principle, this is not a problem, since energy equations routinely solve for both advection and diffusion, in the simplest formulation, yet, strong temperature (and, hence, viscosity) gradients affect the performance of some solvers (mg, for instance) as well as some mesh topologies (e.g., mesh refinements). Some information might help the seamless integration of this tool in the numerical codes. Additionally, this comment applies to the offset ridge in figure 3, where strong horizontal temperature gradient result along the offset zones.*

We have added the McKenzie equation, with a short explanation on the implementation in section 2.1.4. As can be seen in the equation, the advection velocity enters as the plate velocity $v$ in the Reynolds number (equation 2).

Concerning the issue of sharp transitions in the initial temperature distribution: this in general not a problem in application codes because thermal diffusion will quickly smoothen such transitions. If necessary because of potential instabilities application codes can first subject the initial temperature field to a small number of timesteps where only thermal diffusion operates to smoothen the temperature field, before starting the convecting flow.

For this reason the GWB does not currently implement diffusion of the temperature field. It is viewed by the authors as not a core task of the GWB, better performed by geodynamic model codes. Futhermore, in the example cases and in the production geodynamic modelling of 3d subduction zones which the authors did, diffusion by the GWB was not needed.

It is technically possible to add a temperature plugin/model to the GWB which computes diffusion locally for every point, but that would have to construct its own local grid and diffuse over that. This could be added if there is a strong demand from users, but it will make the every query to the world builder a lot more expensive. For now, the authors view this as a function which is more logical (and more efficient) to perform within the geodynamic modelling software used by the user.

We agree that the explanation in the paper on this topic can be improved, and have done so in the discussion.

When we added the equation we noticed two problems in the code computing the McKenzie 1970 temperature model. The problems are described in https://github.com/ GeodynamicWorldBuilder/WorldBuilder/pull/125. When the problems were fixed we noticed that the results of the tester had only changed between about a degree and a few tens of degrees. In practice the difference is not noticeable, as is shown in the figure below. We therefore only updated the stand-alone examples which contain subducting plates of section 3.1, and have not re-run the computations. We also took another look at the examples we show and noticed that the 2D subduction examples in section 3.1.1

are very similar to the SEPRAN example in that they both are a ridge with a subducting and overriding plate where the temperatures are defined by a linear temperatures. To be a bit more diverse in the examples we now instead use the adiabatic temperature to define the mantle and lithosphere temperatures instead of a constant temperature.

*[3] More in general, the way lateral variations are handled remains a bit unclear. In principle, one could use blocks where the properties are piecewise constant, yet, this is hardly a natural case and might prompt unwanted inaccuracy in the numerical solution. Perhaps a simple "smoothing" parameter could be considered, as opposed to a proper solution of the (temperature/concentration/material) diffusion equation, which can be done numerically by the preferred code.*

We agree that continuous lateral variations are not well parameterized yet in the GWB, and that it would add a lot of value to the user in making complex model. There are different ideas of how to best do this, but have not yet been implemented. Since there are already some options (with a little bit of work) in which an approximation of it is possible and we feel that the current state of the GWB is already a great improvement over what is available, we believe that these kind of new features and improvements can be added after the publication. For the remark about smoothing see the answer to [2].

*[4] I find the treatment of the tectonic provinces appealing, although I'd recommend some more focus on the subduction zone. From my experience, in both the mechanical and thermo-mechanical approaches, a control on the interface is critical. While non-linearities in the upper plate do reduce the coupling along this interface, this might not happen at the inception of subduction, where the available pull forces are low, whence the use of "weak zones" to ease subduction. Indeed, this might not be the case of the model in fig.5, yet in three-dimension, where computational cost is at a premium, allowing alternative strategies is important. Likely a back arc zone can be imposed with*

*a thinned area with a piecewise constant thickness, although a more flexible strategy is in order, here*

To reduce the coupling one can add a compositional layer at the top of the slab and tune its rheology(e.g. Quinquis et al, tectonophysics 497, 2011). If one want to have a controllable layer on the other side, a fault layer with the desired properties can be added before the slab is added.

Besides defining a piecewise constant thickness, one can also use the subducting plate, set it to an adiabatic temperature and let it start at a certain depth to carve out the complex 3d shape at the bottom of the lithosphere you want. But this is indeed an area where there is still a lot of room for improvement (like in the answer to question [3]).

*[5] Last, but not least. I find that this tool's great potential likely resides in the embedding of realistic datasets. Perhaps the Authors have some example or some idea on workflows using datasets, such as slab 2.0, populating the thermal field in a slab defined by the Benioff zone, or perhaps embedding oceanic lithosphere age dataset into the ridge model, for instance.*

We agree that this is one of the future features which could add a lot of value to the GWB. We did actually investigate slab 2.0, but we found that we were missing information in the dataset such as distance from the top of the slab and distance from the beginning of the slab. We also found that not all slabs where present or complete in this dataset. This may of course be resolved in the future for this dataset. We are also very interested in adding ways to use tomography datasets to define temperature in the mantle although we wish to proceed here with caution before releasing such features.

Please also note the supplement to this comment:
https://www.solid-earth-discuss.net/se-2019-24/se-2019-24-AC1-supplement.pdf

[Figure]

[Figure]

**Fig. 1.** Temperature field with old McKenzie temperature model implementation for figure 1 in unrevised paper.

[Figure]

**Fig. 2.** Temperature field with new McKenzie temperature model implementation for figure 1 in unrevised paper.

**Supplement:**

[revised manuscript text omitted]

---

## Author Comment (AC2) · 10 Jul 2019

We would like to express our gratitude to the reviewer for the very careful reading ofour manuscript. Peer review makes scientific articles better, and this applies to this oneas well – thank you!

The revised paper has been added as a supplement.

*This paper presents a tool, the Geodynamic World Builder, allowing the simulation of more realistic tectonic features, e.g., a continental, an oceanic or a subducting plate. The paper explains the philosophy of the tool and the definition of different tectonic settings. The second part focuses on 2D/3D cartesian and spherical examples of mid-ocean ridges and subduction zones models. I appreciate the effort made to simulate*

[Figure]

*different tectonic context. I have no doubt this tool will be very beneficial and readily usable by the community. The GitHub documentation is also abundant, including a documentation section and a manual.*

We appreciate the kind words!

**General comments:**

*[1]Basics information, which could benefit potential users, are missing. The models' parameters are not specified, such as the name of the codes used to generate the figures.*

We are not sure what the reviewer is missing. We have provided the GWB input files for all the models and have stated in the introduction that the World Builder can create files which can be visualised by Paraview, and note in the acknowledgements that the data visualization has been caried out by Paraview. We did consider adding the ASPECT example input file, but we think that it would be better fitted as a cookbook in the ASPECT repository than in a paper on the GWB.

*[2] The methodology to define complex geometries/polygons could be more explained. For example, figure E1 is useful and could be included in the main text.*

Defining complex polygons is just a matter of adding the points of the polygon to a list, we are not sure what extra information the reviewer wants in the paper. We have added figure E1 to the main text (now called figure 8) and use it in section 3.3. We replaced figure E1 in the appendix with the code related to that example.

*[3] The authors did not discuss the impact of more realistic geometries on the computational time of the models. Indeed, rapid temperature variation between different materials can induce longer calculation times.*

The main author shows in chapter four of his PhD thesis (http://dspace.library.uu.nl/handle/1874/379767), which is a paper in preparation, that this is in practice not a problem. Nonetheless, if needed in some cases, smoothing features could be added to the GWB later, but they would require some careful design to remain efficient. See answer to question [2] of reviewer 1.

**Specific comments:**

*[1] P1, l15: ". . .constrained by boundary conditions, which can be time-dependent, and by initial conditions. . ." you should briefly explain these conditions here and consider adding the parameters of all the models you present in the paper.*

We have changed ". . .constrained by boundary conditions, which can be time-dependent, and by initial conditions. . ." to ". . .constrained by boundary conditions (e.g. velocity, pressure, temperature or heat-flux boundary conditions), which can be time-dependent, and by initial conditions. . ."

We did not add the a description of all the parameters of the computations, because the few runs which are shown are just to showcase that the generated initial conditions can be used in geodynamic models in general. We feel that going in too much detail would dilute the message of the paper. For the readers interested in the parameter values use for real computations with ASPECT can look at chapter four of the main authors PhD thesis (http://dspace.library.uu.nl/handle/1874/379767), where a complete input file is given. If the reviewer and the editor feel strongly about it, we can add the ASPECT input file for that specific run as an appendix.

We have also added the sentence: "These examples are intended to illustrate the ease of use in different codes instead of the physics details of the models shown." to section 3, to emphesise the intent of the examples.

*[2] P2, l12: "that implicitly define volumes to which temperature and composition can*

*be assigned." Rough variations of temperature, so viscosity, are hard to solve: Could you add a function to avoid this issue?*

It depends on what temperature model is assigned within the volume, and most models do more complex temperature distributions than assigning a uniform temperature to the volume. To highlight this we changed the sentence:

"that implicitly define volumes to which temperature and composition can be assigned."

to

"that implicitly define volumes to which temperature and composition models can be assigned."

This doesn't mean that in complex models, no rough variations of temperature may occur. Especially with the McKenzie (1970) equation, the top of the slab is very hot, while the surface and the continental plate are relatively cold. But we have no experienced problems with running these kind of models, as is also shown in this paper and in the PhD thesis mentioned in the previous comment.

Technically it is possible, but it will require careful design and may significantly increase the computation cost depending on the chosen implementation, because the main aim of the GWB is to provide the answer to 'I am a point, in which temperature/material am I?'. Also see the answer to question [2] of reviewer 1.

*[3] P3, l3-4: "but it can be achieved through a sticky air approach, where air is a composition...". Yes for small models, but such an approach is difficult to implement in 3D spherical models because it drastically increases the calculation time.*

We agree that this is not an optimal situation for those kind of models yet. We have some ideas of how we could greatly improve the situation, but the first author would be very interested to discuss with people who actually need this kind of functionality to find the best way of parameterizing these problems.

[Figure]

*[4] P3, l12-14: "This allows for defining an upper and lower mantle and to insert specific volumetric structures such as Large Low Shear wave Velocity Provinces (LLSVPs) at the core-mantle boundary. In the present version these mantle features can be assigned a radially uniform, linear or adiabatic temperature profile." Could you give an example, it is not clear how you can generate such structures?*

We have changed the sentence "This allows for defining a upper and lower mantle and to insert specific volumetric structures such as Large Low Shear wave Velocity Provinces (LLSVPs) at the core-mantle boundary. In the present version these mantle features can be assigned a radially uniform, linear or adiabatic temperature profile." to "This allows for defining a upper and lower mantle and to insert specific volumetric structures such as Large Low Shear wave Velocity Provinces (LLSVPs) at the core-mantle boundary in the same way as for example an oceanic plate, but at depth. In the present version these mantle features can be assigned a radially uniform, linear or adiabatic temperature profile."

*[5] P3, l26-30: "Dip angles are linearly interpolated along a segment. The overall direction of slab dip can be to either side of the trench and is selected. . .. varying 3D slab morphology." A figure, like figure E1, could help the reader to understand the method.*

We have include figure E1 in the main text and added a reference to it in the first sentence. We also added a reference to the ASPECT figure as an example for the varying 3D slab.

*[6] P5, paragraph 3: I encourage the authors to focus on open source software such as CitcomS, CitcomCU, Underworld,....*

We agree that this has a large potential, and the first author is very much willing to help the developers of those codes to link the GWB. At the time of writing we have made an issue on the Underworld Github page (https://github.com/underworldcode/underworld2/issues/393) to see whether there is interest from that community. The response from the developers has been very positive. Although CitcomS and CitcomCU have official repositories, the actual use of the code is much more decentralized. We feel that adding it to one of the official repositories would not necessarily result in it being available to many Citcom users. We think that helping individual groups who use their own version of Citcom to couple that to the World Builder would be more effective and time efficient. Again, the first author is very willing to help those groups, or any other group with a different code, to carry out the coupling.

*[7] P5, l17: "The slab temperature is computed using the McKenzie model for a particular slab history." I understand this paper is not on geodynamic interpretations, but it could help the reader to add the model parameters.*

We have added the equation (see equation 1) and described the parameters used for the computation of the McKenzie model. The values of the parameters can be found in the world builder files in the appendices, or for the default values in the manual.

*[8] P9, l1-2/Fig. 5: "One sided subduction is obtained in a self-consistent way by the presence of a weak crustal layer of uniform viscosity $10^{21}$ Pa s ËŽus on top of the subducting lithosphere." Is it a self-consistent slab?*

In response to the reviewers comment the model description has been rephrased to clarify the role of the weak crustal layer.

The sentence: "One sided subduction is obtained in a self-consistent way by the presence of a weak crustal layer of uniform viscosity $10^{21}$ Pa s on top of the subducting lithosphere." is replaced by "Subduction is driven in a selfconsistent way by the ridge push resulting from the thickening of the oceanic plate and the negative buoyancy of the subducted slab. Free slip impermeable boundary condition are imposed on the flow. The top of the subducting lithosphere consists of weak crustal layer, 10 km thick and with a uniform viscosity of $10^{20}$ Pa s. This weak crustal layer plays an essential role in preventing the locking of the subducting lithosphere with the overriding plate that would stop the subduction process (Androvičová et al., 2013)." NOTE: the crustal viscosity value has been corrected with the new value $10^{20}$ Pa s.

*[9] P8-9, Paragraph 3.2 to 3.4: the same comment than before: In order to foster the development of open source tools, it could be relevant to add open source software such as CitcomS, CitcomCU, Underworld,..*

We completely agree that this would be very useful and we are we are very much willing to help those open source communities to implement the coupling if there is interest from them. See answer to question 7.

Please also note the supplement to this comment:
https://www.solid-earth-discuss.net/se-2019-24/se-2019-24-AC2-supplement.pdf

**Supplement:**

[revised manuscript text omitted]